# FLUENTLIP: A PHONEMES-BASED TWO-STAGE AP-PROACH FOR AUDIO-DRIVEN LIP SYNTHESIS WITH OPTICAL FLOW CONSISTENCY

## ABSTRACT

Generating consecutive images of lip movements that align with a given speech in audio-driven lip synthesis is a challenging task. While previous studies have made strides in synchronization and visual quality, lip intelligibility and video fluency remain persistent challenges. This work proposes FluentLip, a two-stage approach for audio-driven lip synthesis, incorporating three featured strategies. To improve lip synchronization and intelligibility, we integrate a phoneme extractor and encoder to generate a fusion of audio and phoneme information for multi-modal learning. Additionally, we employ optical flow consistency loss to ensure natural transitions between image frames. Furthermore, we incorporate a diffusion chain during the training of Generative Adversarial Networks (GANs) to improve both stability and efficiency. We evaluate our proposed FluentLip through extensive experiments, comparing it with five state-of-the-art (SOTA) approaches across five metrics, including a proposed metric called Phoneme Error Rate (PER) that evaluates lip pose intelligibility and video fluency. The experimental results demonstrate that our FluentLip approach is highly competitive, achieving significant improvements in smoothness and naturalness. In particular, it outperforms these SOTA approaches by approximately **16.3%** in Fréchet Inception Distance (FID) and **35.2%** in PER.

## 1 INTRODUCTION

Audio-driven lip synthesis, also known as Talking Face Generation (TFG), generates a coherent sequence of mouth movements that are consistent with the given audio input. It has become a prominent topic of research (Jamaludin et al., 2019) due to its wide range of real-world applications, such as film dubbing (Kim et al., 2018), video bandwidth reduction (Suwajanakorn et al., 2017) and face animation (Song et al., 2019). Despite its potential, achieving perfect lip synchronization remains a significant challenge. Hence, it has attracted considerable attention from researchers.

Numerous methods have been proposed to improve synchronization and visual quality in audio-driven lip synthesis. To enhance synchronization, Wav2Lip (Prajwal et al., 2020) extends Sync-Net (Chung & Zisserman, 2017) to the RGB space, using a lip sync discriminator to calculate sync loss and penalize asynchronous lip pose generated. SyncTalkface (Park et al., 2022) calculates sync loss by measuring lip pose feature distances between synthesized and ground truth videos. Talk-Lip (Wang et al., 2023a) leverages a pre-trained lip-reading expert (Shi et al., 2022) to guide lip pose synthesis. Moreover, many approaches, like Wav2Lip, incorporate Generative Adversarial Networks (GANs) (Goodfellow et al., 2014) to enhance visual quality.

While synchronization and visual quality have been well-studied, less attention has been given to improving lip pose intelligibility and video fluency. Notably, TalkLip employs a lip-reading expert to enhance lip pose intelligibility (Wang et al., 2023a). In our work, we address these gaps by proposing a phoneme-based two-stage approach with optical flow consistency (denoted as FluentLip), specifically designed to improve both lip pose intelligibility and video fluency.

Specifically, we utilize a phoneme extractor to automatically recognize and align phonemes from the audio and a phoneme encoder to generate the corresponding phoneme embeddings. These embeddings are then fused with audio embeddings to serve as the reference input for the lip sync

discriminator and generator within GANs. To further enhance fluency, we introduce an optical flow consistency loss that penalizes unnatural transitions between frames during training. Additionally, we employ a diffusion model (Ho et al., 2020) with a diffusion chain to accelerate convergence and stabilize the training process (Wang et al., 2023b), ultimately improving visual quality.

The main contributions of this work are summarized as follows.

- We leverage a phoneme extractor and encoder to create a fusion of phoneme and audio embeddings, improving lip pose intelligibility. Additionally, we develop an optical flow consistency loss to guide training, ensuring smooth transitions between frames and enhancing the naturalness of synthesized videos. Our approach specifically addresses the underexplored challenges of lip pose intelligibility and video fluency in audio-driven lip synthesis.

- We integrate a diffusion chain into the training process of GANs, leading to faster convergence and enhancing the stability of the training process. The quality of the synthesized videos is improved, providing realistic and visually appealing outputs that align closely with the corresponding audio input.

- We evaluate the effectiveness of our proposed FluentLip with five state-of-the-art (SOTA) approaches, demonstrating a notable performance of approximately 16.3% in Fréchet Inception Distance (FID) and 35.2% in Phoneme Error Rate (PER). Additionally, we introduce a novel metric that leverages insights from the lip-reading expert and the Grapheme-to-Phoneme (G2P) model to assess the perceptual performance of various approaches.

## 2 RELATED WORK

### 2.1 SPEECH-DRIVEN TALKING FACE GENERATION

Talking face generation was first proposed in the 1990s (Yehia et al., 1998), with early approaches primarily using Hidden Markov Models (HMM) (Bregler et al., 1997). In recent years, deep learning has emerged as the dominant TFG method, which can be generally classified into intermediate representation-based approaches and reconstruction-based approaches (Park et al., 2022).

The intermediate representation-based approaches focus on learning facial representations, such as 3D meshes, which are used for facial synthesis. For example, SadTalker (Zhang et al., 2023) generates 3D-aware face renders for synthesizing talking faces, while Everybody's Talkin' (Song et al., 2022) reconstructs 3D meshes from extracted facial parameters to generate video sequences. However, these approaches are limited in their generalizability to arbitrary characters, and 3D modeling often struggles to represent mouth details (Wang et al., 2023a).

In contrast, reconstruction-based approaches primarily rely on end-to-end encoder-decoder architectures, which avoid the limitations of intermediate representations and offer improved mouth details synthesis. It began with ObamaNet (Kumar et al., 2017), which focused on a specific character. This was followed by Speech2Vid(Jamaludin et al., 2019) and LipGAN (KR et al., 2019), which improved generalizability allowing for video generation of arbitrary characters. A breakthrough came with Wav2Lip (Prajwal et al., 2020), which introduced SyncNet (Chung & Zisserman, 2017) as a lip sync expert, achieving SOTA synchronization performance. More recent efforts have aimed at improving visual quality based on the Wav2Lip model. Gupta et al. (2023) pre-trained a VQGAN model (Esser et al., 2021) to train Wav2Lip in quantized space, improving visual quality to a maximum of 4K resolution. Diff2Lip (Mukhopadhyay et al., 2024) uses a diffusion model (Ho et al., 2020) to replace the Seq2Seq framework in Wav2Lip, further improving visual quality.

At the same time, some works have taken an alternative approach by focusing on issues that impact human viewing, particularly by integrating generated videos into real-life scenarios. Among the most novel concerns is lip pose intelligibility. Wang et al. (2023a) is the first to highlight this issue in the context of TFG, introducing AV-HuBERT (Shi et al., 2022) as a lip-reading expert to improve lip pose intelligibility and opening a new direction for TFG research.

Despite the increasing number of works in TFG, surprisingly little attention has been given to video fluency. Although these subtle details may be difficult to perceive with the naked eye, as visual quality continues to improve, fluency will become a critical issue. To fill this gap, our work intro-

duces phonemes, commonly used in the Text-to-Speech (TTS) domain, along with a novel metric to promote assessing lip pose intelligibility. Furthermore, we incorporate optical flow consistency loss to improve the fluency of generated videos.

## 2.2 Phoneme-Based Multimodal Learning

Most previous TFG studies employ audio or text as the input driver with their unimodal learning model. ATVGnet (Chen et al., 2019) and Wav2Lip (Prajwal et al., 2020) employ audio as driven input, while ParaLip (Liu et al., 2022) and Make-A-Video (Singer et al., 2023) are text-driven. Since audio varies with different speakers and text may contain homophones, it's difficult to represent speech content with just one of them. Phoneme (Zhang et al., 2022) is a more microscopic concept widely used in the TTS domain, focusing on syllables rather than words. Although some previous works employ phonemes in TFG, such as Text2video (Zhang et al., 2022) and text-based editing video (Fried et al., 2019), few of them have extended unimodal to multimodal learning. Thus, we introduce multimodal learning in our work by combining audio and phoneme as driven inputs. Phonemes capture precise speech content, while audio conveys robust and ample information, helping to judge speech content accurately and ultimately enhancing lip pose intelligibility.

## 2.3 Consecutive Image Generation

Video generation can be viewed as a process of generating consecutive images in frames, and the three primary frameworks are prevalent for solving it: Seq2Seq, GANs (Goodfellow et al., 2014), and diffusion model (Ho et al., 2020). Among these, Seq2Seq serves as the foundational model, while diffusion models have demonstrated outperforming GANs in image generation (Dhariwal & Nichol, 2021). Most of the previous TFG works use Seq2Seq to generate frame images, often coupled with GANs to enhance the visual quality of these images, such as Wav2Lip (Prajwal et al., 2020). Some approaches use the diffusion model as an image generator, which also achieves good results, such as Diff2Lip (Mukhopadhyay et al., 2024). Nevertheless, GANs training often comes up with the mode collapse (Wang et al., 2023b), a challenge that has been overlooked in previous TFG works that leverage GANs. To mitigate this, Wang et al. (2023b) propose Diffusion-GAN, which integrates a diffusion model into the GANs training to generate Gaussian instance noises in high-dimensional data space, effectively improving the stability and overall performance of GANs training. Inspired by it, our work also employs a diffusion model to stabilize GANs training and improve its performance.

Unlike naive image generation, consecutive image generation should consider the fluency between frames. In real-life videos, objects are moving regularly with specific trends, so the pixel points move more smoothly. Naive image generation methods often neglect this, leading to irregular and trendless pixel point movement between frames. Optical flow (Horn & Schunck, 1981), a technique frequently used to measure pixel displacement between two consecutive frames, is estimated by FlowNet (Dosovitskiy et al., 2015; Ilg et al., 2017) or more popular Recurrent All-pairs Field Transforms (RAFT) (Teed & Deng, 2020) and is often applied in dynamic image detection. For example, self-driving automobiles use optical flow to predict the motions and traces of surrounding objects (Hu et al., 2020). Therefore, we assert that optical flow is an effective measure of video fluency and design a novel loss function based on optical flow to penalize irregular pixel moves generated, enhancing video fluency. Note that our work first employs optical flow in the TFG domain.

## 3 The Proposed FluentLip Approach

The two-stage approach that combines a lip sync discriminator and a lip synthesis network has proved to be quite successful for audio-driven lip synthesis (Prajwal et al., 2020; Gupta et al., 2023; Mukhopadhyay et al., 2024). Leveraging this powerful framework, we design dedicated fused embedding and optical flow consistency strategies to address lip pose intelligibility and video fluency, and to improve lip synchronization.

Algorithm 1 outlines the architecture of FluentLip, which adopts a two-stage process. In stage 1, phonemes are automatically extracted from the audio corresponding to a given video frame, and aligned precisely by frame. A phoneme encoder generates phoneme embeddings, which are then

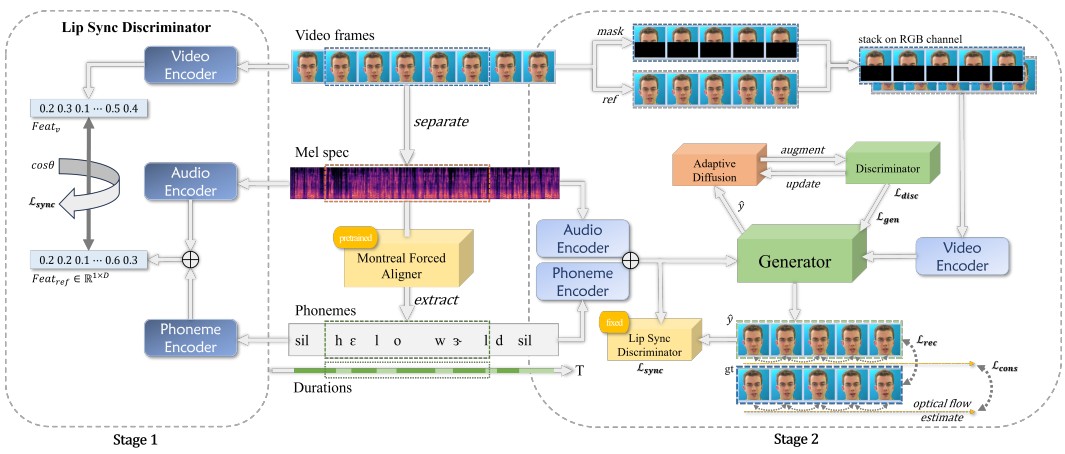

Figure 1: The architecture of the proposed FluentLip approach

fused with the audio embeddings and fed into the lip sync discriminator alongside the corresponding video embeddings. This fusion of sensory modalities establishes multimodal learning.

In stage 2, the fused audio and phoneme embeddings are used to train the lip generator, together with video embeddings from stacked frames of both predicted and reference images. The synthesized facial video is guided by the fixed lip sync discriminator from stage 1 via a sync loss to ensure precise lip synchronization, and by the visual discriminator of GANs to improve visual quality.

Additionally, an adaptive diffusion model is employed between the generator and visual discriminator, where a diffusion chain of variable length is applied to gradient propagation to improve the stability and effectiveness of the training process. To further improve the realism of the synthesized video, the RAFT model predicts optical flow between frames, applying optical flow consistency loss to penalize unnatural shifts. All losses are integrated to optimize the training of the whole network. Below, we provide a detailed description of each core component of the FluentLip approach.

### 3.1 STAGE 1: LIP SYNC DISCRIMINATOR

**Phoneme encoder** The phoneme encoder is to effectively encode the phoneme sequence, which is subsequently concatenated and fused with the audio embedding. The raw phoneme text and its corresponding durations are automatically extracted from the reference audio by a pre-trained phoneme extractor Montreal Forced Aligner (MFA) (McAuliffe et al., 2017), as illustrated in Fig. 1. The phoneme text sequence is first converted into numerical representations via a global phoneme table. Given that the length of the phoneme sequences varies across different audio clips, we pad both the phoneme encodings and their corresponding duration vector to a fixed length before proceeding with the embedding process. Within the phoneme encoder, positional encoding is employed alongside a Transformer-based architecture, which improves the model's ability to capture the sequential dependencies inherent in the phonemes, ultimately generating high-quality phoneme embeddings.

Let us denote $V_{raw} \in \mathbb{R}^x$ and $L_{raw} \in \mathbb{R}^x$ as the initial phoneme encoding and duration vector respectively, where $x$ is the irregular length of each phoneme vector. The padded phoneme encoding and duration vector are denoted as $V_{pad} \in \mathbb{R}^T$ and $L_{pad} \in \mathbb{R}^T$, where $T$ is the fixed phoneme vector length. The embedded phoneme vector is represented as $V \in \mathbb{R}^{T \times D}$, where $D$ is the feature dimension, and $L \in \mathbb{R}^T$ is the normalized duration vector. Additionally, $P \in \mathbb{R}^{T \times D}$ denotes the positional encoding vector. The sequential concatenation of $V$, $L$ and $P$, denoted as $V_{cat} \in \mathbb{R}^{T \times (D+1+D)}$, is fed to the Transformer network, which processes the input to generate the penultimate phoneme embedding $V_{tm} \in \mathbb{R}^{T \times (D+1+D)}$. Subsequently, a linear layer followed by a batch normalization layer produces the ultimate phoneme embedding $Y \in \mathbb{R}^{T \times (D \times 2)}$, which serves as the output. The whole procedure of phoneme encoding is shown in Fig. 2.

**Lip sync discriminator** The lip sync discriminator, such as the previously proposed Sync-Net (Chung & Zisserman, 2017), aims to evaluate the synchronization between a Mel spectrum clip

Figure 2: The phoneme encoding procedure, with $t$ representing the unit time for duration.

and a lip motion clip by comparing their embeddings under a latent space. Inspired by Wav2Lip (Prajwal et al., 2020), which first applied the lip sync discriminator into the TFG to improve lip synchronization, we also integrate this module, introducing phonemes as a novel addition. The cosine similarity between the audio and video feature vectors is calculated and used to obtain the sync loss by computing Binary Cross-Entropy (BCE).

Let us denote $y$ as the target similarity, whose value reflects whether the audio-video pair is originally matched, and $S(m, n)$ is the cosine similarity function for feature vectors $m$ and $n$. For $N_i$ audio-video pairs with audio embedding $a$ and video embedding $v$, the sync loss is formulated as:

$$\mathcal{L}_{sync} = \frac{1}{N_i} \sum_{i}^{N_i} [-y_i \log S(a_i, v_i) - (1 - y_i) \log (1 - S(a_i, v_i))] \tag{1}$$

We fuse the phoneme embedding, extracted from the audio and encoded by the phoneme encoder, with the original audio embedding. This fused embedding replaces the audio-only embedding for training the lip sync discriminator. Considering the Mel spectrum varies significantly across speakers and even across sentences from the same speaker, phonemes are a relatively stable feature that is consistent as long as the speech content remains the same, regardless of speaker or style. Thus, combining audio with phonemes leads to more accurate lip sync guidance and more stable lip motion synthesis. Denoting the phoneme embedding as $p$, the ultimate sync loss is formulated as:

$$\mathcal{L}'_{sync} = \frac{1}{N_i} \sum_{i}^{N_i} [-y_i \log S(a_i + p_i, v_i) - (1 - y_i) \log (1 - S(a_i + p_i, v_i))] \tag{2}$$

Once training stage 1 is done, the lip sync discriminator is fixed and serves as guidance for training stage 2. In this stage, the sync loss penalizes the mismatched motion of synthesized lips by comparing the video with the fused audio-phoneme reference, prompting the lip generator to produce more realistic, synchronized, and fluent lip image frames.

### 3.2 STAGE 2: LIP SYNTHESIS

**Lip synthesis networks** Similar to previous studies (Prajwal et al., 2020; Wang et al., 2023a; Gupta et al., 2023), our lip generator uses a Seq2Seq network for reconstruction, comprising two audio and video encoders with an additional phoneme encoder and one decoder. Like the lip sync discriminator, the generator processes a triple tuple input, including an audio clip, a phoneme sequence with durations, and an image frame. The image frame is stacked on the RGB channels with two images from a video clip, one randomly selected as a full identity reference while the other with its

lower half masked to predict the lip pose. The audio and visual elements are fed into a CNN-based encoder, while the phoneme sequence and duration are processed by the Transformer-based encoder described in Sec. 3.1, generating three embeddings. These embeddings are combined and passed to the CNN-based decoder, which generates the output layer by layer. Finally, the predicted face is separated from the stacked image and applied to the original video. The objective is to synthesize a facial image that closely resembles the original face but with the lip driven by the reference audio and phonemes. Given $N_i$ pairs of synthesized facial images $v^{'}$ and ground truth images $v$, we adopt L1 loss as the reconstruction loss between the synthesized and real facial images, which is calculated as:

$$\mathcal{L}_{rec} = \frac{1}{N_i} \sum_{i}^{N_i} |v_i - v_i^{'}| \tag{3}$$

In our lip synthesis networks, the synthesized images are augmented with noise through an adaptive diffusion model before being inputted to the visual discriminator, equivalent to the discriminator of GANs. This diffusion chain is a novel approach shown to improve training stability and efficiency of the GANs, which will be introduced in the following subsection.

Let us denote $D$ as the visual discriminator in Fig. 1. The generator and discriminator losses caused by the visual discriminator are defined as:

$$\mathcal{L}_{gen} = \frac{1}{N_i} \sum_{i}^{N_i} \text{-} \log(1 - D(v^{'})) \tag{4}$$

$$\mathcal{L}_{disc} = \frac{1}{N_i} \sum_{i}^{N_i} [\text{-} \log D(v) - \log(1 - D(v^{'}))] \tag{5}$$

The generator loss $\mathcal{L}_{gen}$ propagates gradients back to improve the quality of synthesized facial images, while the discriminator loss $\mathcal{L}_{disc}$ strengthens the ability of discriminator to distinguish between synthesized and real facial images. Together, these losses drive the mutual reinforcement of the GANs.

**Optical flow consistency loss** The optical flow consistency loss is commonly used in stereo matching (Lai et al., 2019) and multi-view stereo tasks (Furukawa et al., 2015), comparing luminance consistency and motion smoothness between consecutive frames. Considering our task is to generate continuous image frames for fluent video output, we adopt the optical flow consistency loss as part of the total guidance of the generator to penalize anomalous motion variations among synthesized facial images. Unlike previous works such as Wav2Lip, which simply focus on audio-video consistency, our approach also ensures video inter-frame consistency. This is especially important when the original video features significant motion, with frequent changes in lip angle and pose.

To calculate this loss, we estimate the optical flow consistency between synthesized and real image sequences using a pre-trained RAFT model (Teed & Deng, 2020), applying L1 loss as the optical flow consistency metric. Let $F(m, n)$ represent the optical flow estimating function for two dynamic images, $m$ and $n$. Given $N_i$ pairs of synthesized facial images $v^{'}$ and ground truth images $v$, the optical flow consistency loss is defined as:

$$\mathcal{L}_{cons} = \frac{1}{(N_i - 1)} \sum_{i=2}^{N_i} |F(v^{'}_i, v^{'}_{i-1}) - F(v_i, v_{i-1})| \tag{6}$$

Finally, the total loss for optimizing the lip synthesis networks combines all the aforementioned loss components and is formulated as follows:

$$\mathcal{L}_{total} = \lambda_{sync} \cdot \mathcal{L}^{'}_{sync} + \lambda_{rec} \cdot \mathcal{L}_{rec} + \lambda_{gen} \cdot \mathcal{L}_{gen} + \lambda_{cons} \cdot \mathcal{L}_{cons} \tag{7}$$

where $\lambda_{sync}$, $\lambda_{rec}$, $\lambda_{gen}$ and $\lambda_{cons}$ are scale factors that adjust the contributions of loss components.

**Adaptive diffusion model** Inspired by Diffusion-GAN (Wang et al., 2023b), which proposes a Gaussian mixture distribution over all diffusion steps in a forward length-adaptive diffusion chain to improve the stability and efficiency of GANs training, we integrate a similar technique into our framework. While maintaining the original GANs, we employ an additional diffusion model to

noise-augment the facial images fed into the discriminator. This leads to enhanced training performance, as the generator benefits from its gradients backpropagating through the forward diffusion chain. The chain's length is adaptively adjusted by controlling the noise proportion added to both synthesized and real facial images, based on the discriminator's performance.

The integration of adaptive diffusion model between the generator and discriminator will be demonstrated to accelerate convergence and stabilize the training process in Sec. 4.3, marking a successful practice of injecting instance noise in lip synthesis tasks.

# 4 EXPERIMENTS

## 4.1 EXPERIMENTS SETTINGS

**Dataset** We train our model using the LRS2 dataset (Afouras et al., 2018) and evaluate it on unseen test sets of both the GRID (Cooke et al., 2006) and LRS2 datasets. The GRID is a large multi-talker audiovisual sentence corpus whose video files have a resolution of 720×576 and a frame rate of 25 fps. The audio from each video file is extracted with a maximum amplitude value of 1 and downsampled to 16 kHz. Sentences of GRID consist of a relatively fixed length of independent short words. The LRS2 is an open-world audio-visual speech recognition dataset whose video and extracted audio files have the same parameters of 25fps and 16kHz with GRID, respectively. Unlike GRID, sentences of LRS2 have more meaningful content of varying lengths, and the scenes are more diverse and irregular, making LRS2 more reflective of real-life scenarios.

**Metrics** We evaluate lip synchronization and the quality of synthesized images using widely used metrics such as FID (Heusel et al., 2017), SSIM (Wang, 2004), LSE-D and LSE-C (Prajwal et al., 2020). LSE-D and LSE-C are calculated via a pre-trained sync net to measure the synchronization of lip movements, while FID and SSIM quantitatively assess image quality. In addition, we proposed a novel metric called Phoneme Error Rate (PER), which evaluates lip pose intelligibility and video fluency. PER is computed by comparing phonemes predicted from synthesized video with real phonemes extracted from audio, using a pre-trained lip-reading model AV-HuBERT (Shi et al., 2022). Unlike the Word Error Rate (WER) metric proposed by AV-HuBERT and adopted by Talk-Lip (Wang et al., 2023a) in TFG, PER focuses directly on phonemes, avoiding the shortcomings of word-based evaluation, as the same phoneme sequence can represent multiple distinct words.

**Baselines** We compare our model against several SOTA lip synthesis models, including ATVGnet (Chen et al., 2019), Wav2Lip (Prajwal et al., 2020), SadTalker (Zhang et al., 2023), Talk-Lip (Wang et al., 2023a), and Diff2Lip (Mukhopadhyay et al., 2024). ATVGnet is the first model to use an Attention-based Transformer Network (AT Network) and a Video Generator Network (VG Network) for generating talking face videos. Wav2Lip introduced the innovative use of a lip sync net in its reconstruction-based method. SadTalker generates videos by leveraging intermediate 3D Morphable Models (3DMM) and a 3D-aware face renderer. TalkLip builds upon Wav2Lip by integrating lip-reading loss and contrastive loss with guidance from a lip-reading expert. Diff2Lip is the latest SOTA model, adopting a diffusion model instead of the traditional Seq2Seq framework, achieving superior performance in lip synthesis.

**Implementation Details** We have trained our models in environment configuration as follows: OS of Ubuntu20.04, CPU of AMD EPYC 9754 (18v CPU), GPU of RTX4090D (24GB) and RAM of 60GB. Our model has been trained in stage 1 for 90k steps with a batch size of 40, and in stage 2 for 35k steps with the same batch size. To ensure fairness and rigor, we carry out the experiments for both our model and other approaches under the same setting and on a consistent range of the dataset.

## 4.2 EXPERIMENTAL RESULTS

**Quantitative results** The performance comparison of lip synchronization and visual quality of synthesized images of different approaches on the metrics mentioned above is shown in Tab. 1 for both the GRID and LRS2 datasets. Guided by the diffusion model and optical flow consistency loss, FluentLip achieves near SOTA performance in terms of visual quality, realism and video fluency, with excellent synchronization. Our FluentLip attains top scores in FID, SSIM and PER, while also achieving competitive scores in LSE-D and LSE-C, which reflect lip synchronization.

Table 1: Quantitative performance comparisons of six different approaches on GRID and LRS2 datasets. PER is excluded from GRID due to the lack of semantic content in its sentences, making lip-reading predictions unreliable.

| Methods | GRID | | | | LRS2 | | | | |
|---|---|---|---|---|---|---|---|---|---|
| | LSE-D↓ | LSE-C↑ | FID↓ | SSIM (%)↑ | LSE-D↓ | LSE-C↑ | FID↓ | SSIM (%)↑ | PER (%)↓ |
| Ground Truth | 7.213 | 6.143 | 0.00 | 100.00 | 6.252 | 10.427 | 0.00 | 100.00 | 76.83 |
| ATVGnet | 7.081 | 5.523 | 36.00 | 90.35 | 6.109 | 8.323 | 29.36 | 83.76 | 90.56 |
| Wav2Lip | 6.352 | 6.627 | 26.71 | 96.10 | 5.487 | 11.516 | 28.80 | 91.93 | 77.92 |
| SadTalker | 7.195 | 5.542 | **20.08** | 87.80 | 5.524 | 9.792 | 98.50 | 55.59 | 73.91 |
| TalkLip | 5.808 | **7.534** | 35.38 | 95.85 | 5.755 | 10.561 | 22.71 | 92.64 | 47.31 |
| Diff2Lip | **5.710** | 6.903 | 33.70 | 95.37 | **4.748** | 11.926 | 19.83 | **94.54** | 82.93 |
| **FluentLip** | 6.258 | 6.790 | **21.94** | **96.25** | 5.018 | **11.984** | **16.93** | 93.31 | **46.91** |

FluentLip ranks second in LSE-D and first in LSE-C on the LRS2 dataset, highlighting the effectiveness of our phoneme-based multimodal learning strategy for improving synchronization. Moreover, FluentLip's standout performance in PER on the LRS2 dataset, especially when compared to models without the guidance of a lip-reading expert, underscores the model's superior lip pose intelligibility and its accurate alignment between audio and lip movements. This further confirms the strength of our phoneme-based strategy. The performance on the GRID dataset, which is less varied in terms of background and speech content compared to LRS2, still shows FluentLip's strengths in synchronization, as evidenced by its high LSE-D and LSE-C scores. FluentLip also demonstrates strong visual quality with leading FID and SSIM scores, reflecting its generalizability across unseen datasets.

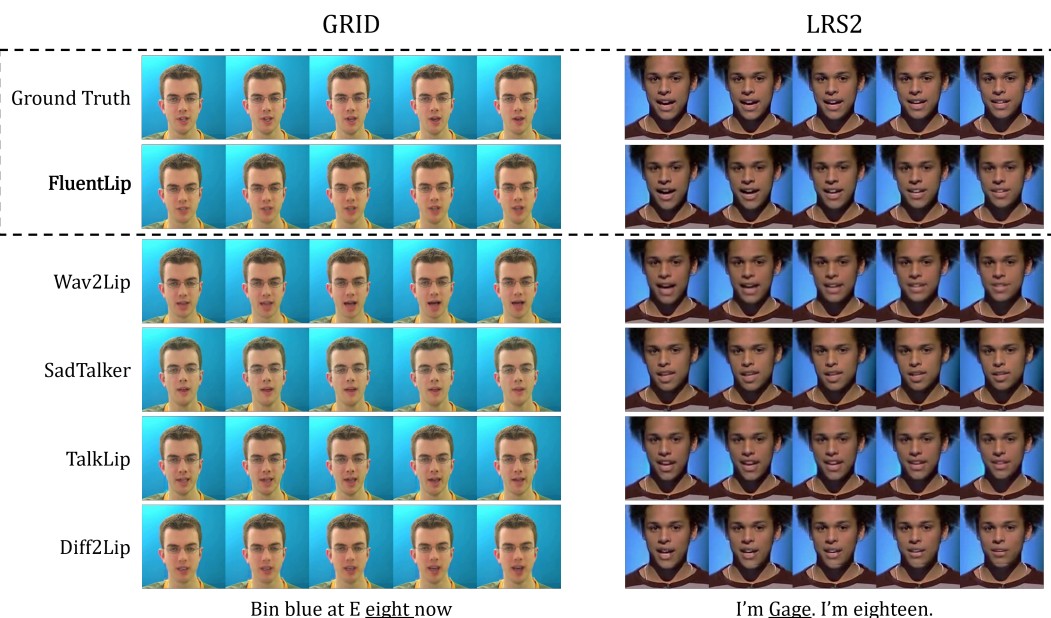

Figure 3: Qualitative comparison on five consecutive frames of the video from different approaches

Both FID and SSIM are metrics that measure similarity between images, but in this case, we apply them to videos. FID evaluates the similarity in aspects such as visual quality, head motion trends, and lip poses, making it a comprehensive metric of video quality, synchronization and fluency. FluentLip's FID is second only to SadTalker's on the GRID dataset and outperforms others on the LRS2 dataset, showing the highly competitive performance of FluentLip in balancing visual quality, synchronization, and fluency. SSIM, which directly measures image realism, places FluentLip ahead of Wav2Lip and TalkLip on the GRID dataset, and just slightly behind Diff2Lip on the LRS2 dataset, showing FluentLip's robustness in producing realistic video. It is worth noting that SadTalker, which

generates facial animations from a single static image rather than a consecutive video, performs differently on the more static GRID dataset and on the more dynamic LRS2 dataset. As a result, SadTalker's performance is optimized for datasets with fewer facial motions and expression changes, whereas FluentLip excels in handling more dynamic content like that found in LRS2.

**Qualitative results** To qualitatively compare the videos generated by FluentLip with those generated by the other models, we present Fig. 3, which shows five consecutive frames from the videos generated by FluentLip and the different models using two arbitrarily selected videos and their corresponding audio from the test set as input. Specifically, the first row displays the ground truth video, and the second row shows the video frames generated by FluentLip, followed by the video frames generated by the other models in sequence. Moreover, we select and zoom in on a single lip pose from each image in Fig. 3 to demonstrate differences in lip poses between FluentLip and the other models more closely, as shown in Fig. 4.

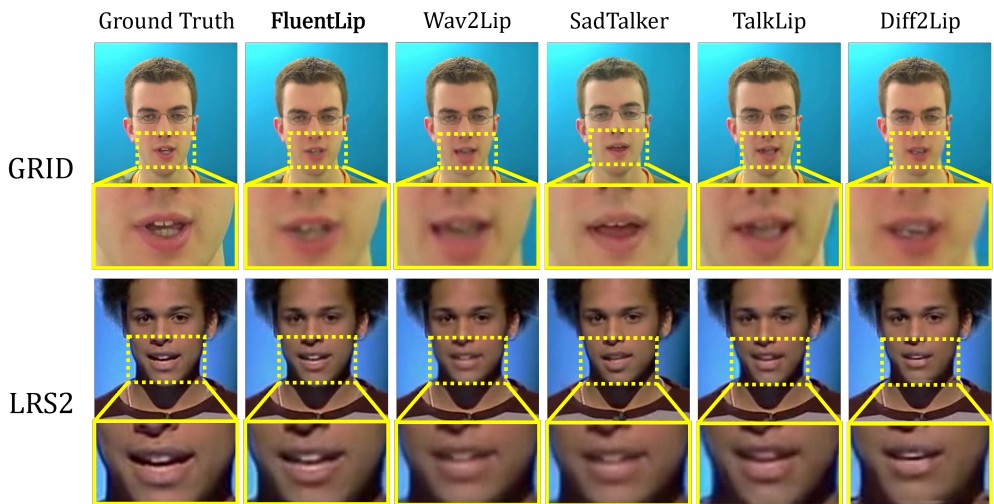

Figure 4: Single frame picked from Fig. 3 and zoomed in on the lip region

From Fig. 3, it is evident that FluentLip generates the most similar image frames to ground truth video regarding synchronization and smoothness. When compared to TalkLip and Diff2Lip, FluentLip generates highly consistent images with ground truth, without any abnormal color block in the video background. Furthermore, in comparison to Wav2Lip and Diff2Lip, FluentLip generates visible and significantly shaped teeth. Against SadTalker, FluentLip produces clear and natural faces with coherent and synchronized expressions and motions. Note that the five consecutive frames from SadTalker appear almost identical, suggesting that the use of 3D may lead to a static expression for the facial animation.

### 4.3 ABLATION STUDY

To verify the effectiveness of each proposed key component, we have trained two variants of our FluentLip model under the following conditions: (1) without the integration of the optical flow consistency loss (**FluentLip (w/o cons)**), and (2) without the integration of the diffusion model (**FluentLip (w/o diff)**). For fair comparisons, FluentLip and its two variants have undergone the same training process in stage 1. In stage 2, they have been trained for 35,000 steps with a batch size of 40.

**Training results** To intuitively compare the performance of our FluentLip and its two variants during training, we select the variation of several crucial losses as shown in Fig. 5.

First of all, regarding the diffusion model, FluentLip (w/o diff), which disables the diffusion chain during the GANs training, exhibits gradual mode collapse in the medium term. Despite losses getting down quickly at the beginning, the unstable training process leads to poor end results. This is evident in the downward and subsequent upward trend of losses (a), (b) and (c) in Fig. 5. This phenomenon is mainly caused by the repression of discriminator, as shown in Fig. 5 (d). However,

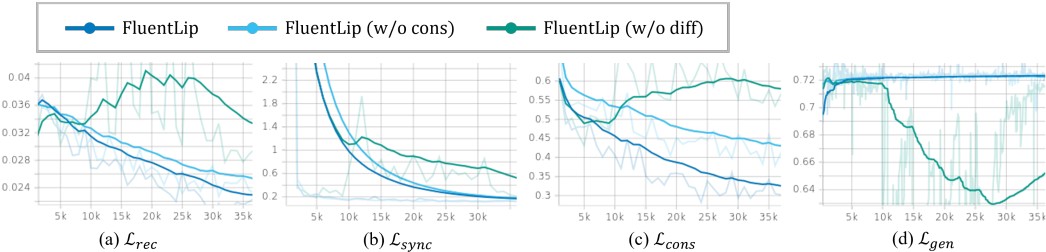

Figure 5: Variation of different losses of FluentLip and its two variants on the training of stage 2

Table 2: Quantitative performance comparisons of FluentLip and its two variants on GRID and LRS2 datasets.

| Methods | GRID | | | | LRS2 | | | | |
|---|---|---|---|---|---|---|---|---|---|
| | LSE-D↓ | LSE-C↑ | FID↓ | SSIM (%)↑ | LSE-D↓ | LSE-C↑ | FID↓ | SSIM (%)↑ | PER (%)↓ |
| FluentLip | 6.258 | 6.790 | 21.94 | 96.25 | 5.018 | 11.984 | 16.93 | 93.31 | 46.91 |
| FluentLip (w/o cons) | 6.825 | 6.485 | 23.13 | 96.22 | 5.629 | 11.206 | 24.60 | 90.54 | 66.25 |
| FluentLip (w/o diff) | 7.594 | 5.917 | 82.48 | 94.37 | 5.048 | 11.928 | 25.81 | 91.20 | 68.37 |

this issue is effectively mitigated by integrating the diffusion model, as demonstrated by FluentLip (w/o cons) and FluentLip.

When evaluating the impact of the optical flow consistency loss, FluentLip (w/o cons) consistently lags behind FluentLip, which utilizes the optical flow consistency loss. This clearly indicates that the optical flow consistency loss is beneficial for generating facial images with higher synchronization and visual quality. Models that adopt this loss function, such as FluentLip, achieve lower reconstruction and synchronization losses.

Overall, these results provide strong evidence that all proposed key components positively influence both the training process and the final outcomes, confirming their effectiveness.

**Quantitative results** We evaluated our FluentLip and two variants on both the GRID and LRS2 datasets using the same metrics as before. The comparisons across different metrics are presented in Tab. 2. As shown in the table, the quantitative performances of the three models generally align with the training results. Specifically, the overall metrics for FluentLip (w/o diff), FluentLip (w/o cons), and FluentLip exhibit a progressively superior trend, with FluentLip achieving the best results overall. Notably, the performance of FluentLip (w/o diff) varies significantly across different datasets, highlighting the instability of GANs, particularly concerning visual quality when the diffusion model is not utilized. The quantitative results, combined with the training findings, demonstrate that each of our proposed key components positively impacts the results, enhancing lip synchronization, visual quality as well as fluency.

## 5 CONCLUSION

In this work, we have studied the challenges inherent in the talking face generation by proposing the FluentLip approach, which synthesizes facial videos with improved fluency and lip pose intelligibility. Unlike previous approaches that primarily focus on synchronization and visual quality, our FluentLip emphasizes lip intelligibility and video fluency by incorporating several novel components. We introduce optical flow consistency loss and utilize phonemes as input to enable multimodal learning, while also employing a diffusion model to stabilize the training of GANs.

Extensive experiments demonstrate the effectiveness of the proposed FluentLip approach, showcasing highly competitive performances in lip synchronization and visual quality compared to five SOTA approaches from the literature. Notably, FluentLip outperforms these approaches in terms of fluency. In addition to these computational results, we conduct an in-depth analysis of the key components to shed light on their roles in the performance of the proposed approach.

ETHICAL STATEMENTS

We certify that this manuscript is original, has not been published, and will not be submitted elsewhere for publication while being considered by ICLR. The study is not split into several parts to increase the number of submissions submitted to various journals or to one journal over time. No data have been fabricated or manipulated (including images) to support our conclusions. No data, text, or theories by others are presented as if they were our own.

In addition, the subject of video synthesis that we are researching may be used by outlaws to negatively impact society. For example, synthesizing videos of controversial speeches of public figures to cause social unrest, synthesizing videos of people around us committing fraud, and infringing on people's portrait rights may also occur. We certify that we will not use this technology for the purposes above, nor will we distribute it to others for non-scientific purposes.

The submission has been received explicitly from all co-authors. Authors whose names appear on the submission have contributed sufficiently to the scientific work and, therefore, share collective responsibility and accountability for the results.

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
