# OpenReview forum: "FluentLip: A Phonemes-Based Two-stage Approach for Audio-Driven Lip Synthesis with Optical Flow Consistency"
_ICLR.cc/2025/Conference — ICLR 2025 Conference Withdrawn Submission_

### Official Review · Reviewer_Hiqn · 2024-10-18

**Soundness:** 2
**Presentation:** 2
**Contribution:** 1
**Rating:** 3
**Confidence:** 5

**Summary:**

The paper introduces FluentLip, which integrates a phoneme extractor and encoder to generate a fusion of audio and phoneme information for multimodal learning, thereby improving lip synchronization and clarity. Optical flow consistency loss is employed to ensure natural transitions between frames. Additionally, the paper incorporates a diffusion chain during the training of GANs to improve stability and efficiency.

**Strengths:**

The talking head generation task addressed in the paper is of practical significance.

**Weaknesses:**

1. The paper does not provide a demo video in the supplementary materials. In a research field that places great emphasis on results, a demo video is crucial. Moreover, since the paper claims video fluency as a contribution, without a demo video, this aspect cannot be verified at all.
2. Limited novelty. The paper does not explain the advantages of phonemes over deep learning features such as wav2vec, and phonemes have already been used in previous works, such as write-a-speaker[1], AVCT[2], etc. This method, when using phonemes, employs an encoder to extract phoneme features, similar to previous approaches [1,2], which is not considered novel. The lip expert mentioned in the paper is also a widely used technique, used in wav2lip[3]. Furthermore, the video fluency of existing methods, such as VASA[4] EMO[5], is already quite good, making the optical consistency loss unnecessary.

[1] Li, Lincheng, et al. "Write-a-speaker: Text-based emotional and rhythmic talking-head generation." Proceedings of the AAAI conference on artificial intelligence. Vol. 35. No. 3. 2021.

[2] Wang, Suzhen, et al. "One-shot talking face generation from single-speaker audio-visual correlation learning." Proceedings of the AAAI Conference on Artificial Intelligence. Vol. 36. No. 3. 2022.

[3] Prajwal, K. R., et al. "A lip sync expert is all you need for speech to lip generation in the wild." Proceedings of the 28th ACM international conference on multimedia. 2020.

[4] Xu, Sicheng, et al. "Vasa-1: Lifelike audio-driven talking faces generated in real time." arXiv preprint arXiv:2404.10667 (2024).

[5] Tian, Linrui, et al. "Emo: Emote portrait alive-generating expressive portrait videos with audio2video diffusion model under weak conditions." arXiv preprint arXiv:2402.17485 (2024).

**Questions:**

Could you demonstrate through experiments the advantages of using phonemes compared to features like wav2vec, and propose a more novel structure for extracting phoneme features? This would enhance the soundness and novelty of the paper.

---

### Official Review · Reviewer_PNdT · 2024-11-03

**Soundness:** 3
**Presentation:** 2
**Contribution:** 3
**Rating:** 5
**Confidence:** 3

**Summary:**

This work proposes a novel two-stage approach for audio-driven lip synthesis, called FluentLip, addressing the challenges of lip intelligibility and video fluency. To enhance lip synchronization and intelligibility, they integrate a phoneme processing module that fuses audio and phoneme information for effective multimodal learning. Additionally, they introduce optical flow consistency loss to ensure natural transitions between image frames. The method also incorporates a diffusion chain during the training of Generative Adversarial Networks, improving both training stability and synthesis efficiency. Extensive experiments conducted on two commonly used datasets show that FluentLip outperforms existing methods, achieving new state-of-the-art performance in lip intelligibility and video fluency.

**Strengths:**

(1)The paper introduces a novel method, FluentLip, for Audio-driven lip synthesis. This approach provides a comprehensive solution to the problem.
(2)The design of the phoneme encoding procedure is reasonable, effectively extracting phoneme information and contributing to the generation of fluent lip movements.
(3) The paper is well-organized and written with a clear structure that facilitates easy navigation through its content.

**Weaknesses:**

(1)The qualitative results presented in the paper are inadequate, as two images take up a significant amount of space while using the same example, limiting the effectiveness of the demonstration. Moreover, the improvements achieved by FluentLip are not clearly evident from the images. Since FluentLip aims to address video fluency and intelligibility, creating a demo page to showcase the generated video results would likely provide a better demonstration of its capabilities.
(2)The three architectural improvements mentioned in the paper seem to lack a close connection. The latter two improvements appear to be mere tricks added to increase workload, rather than being well-integrated components of the overall framework.
(3)The ablation study lacks experiments and analysis regarding the phoneme processing module.

**Questions:**

(1)You proposed a new evaluation metric, the Phoneme Error Rate (PER), which is based on phonemes. Since your method also involves the extraction and processing of phonemes, could you clarify whether this metric is fair in comparison to other methods? Please explain the rationale behind this metric, or provide examples where it demonstrates advantages over Word Error Rate (WER).
(2)In Figure 2, the output of the phoneme encoder is denoted as Y. How is Y combined with the output of the audio encoder to generate Feat_ref in Figure 1, given that their dimensions appear to be mismatched?

---

### Official Review · Reviewer_rVdr · 2024-11-04

**Soundness:** 2
**Presentation:** 2
**Contribution:** 2
**Rating:** 5
**Confidence:** 4

**Summary:**

This work aims to improve lip intelligibility and video fluency in the task of audio-driven lip synthesis. Three points are included to achieve this target: combination of phonemes embedding and audio embedding, introduction of optical flow consistency loss, and incorporation of a diffusion chain in the generation process. Experiments on GRID and LRS2 show the good performance.

**Strengths:**

It is hot and meaningful to explore different aspects of generating high-quality videos to boost the development of this domain. The paper is writing well and it’s easy to follow.

**Weaknesses:**

(1) Several recent work are not compared or included in the paper. Audi-driven generation is a hot topic, and there are several related works in related conferences. But only a single paper of 2024 is included in the work. In addition, there are many works aiming to generate high-quality talking faces using diffusion models driven by audio, such as Hallo, Echomimic, EMO, Loopy and so on, which are all published in 2024. The results of them are high-fidelity of 512*512 and synchronized with audio input. I am not clear about the advantages of this paper compared with those work.
(2) Both the core idea and core framework of this paper are not so insightful. Three contribution points are claimed in the work. The general two-stage framework is popular in this domain and not new here. One contribution claimed is the introduction of phoneme obtained by the pre-processing forced alignment, but there is no ablation study to show the effect of using and not using the phoneme embedding. In addition, the role of phoneme is similar the lipreading or asr part with avhubert, which has been common in previous works. I am not clear about the new role of using phonemes here.
Another contribution claimed is the introduction of optical flow loss with a pretrained RAFT model. But this loss doesn’t contribute enough for an ICLR paper in my idea, because it didn’t bring intrinsic or insightful changes.

**Questions:**

I am not sure about what would be the results if using only audio embeddings with avhubert, without the specific phoneme embedding. Because the features of avhubert could also reflect the phoneme features, what is the intrinsic difference of using avhubert features and using a specific phoneme embedding which has to perform forced alignment preprocessing firstly. If there are no text transcriptions and only audio is provided as input (which is the usual case), I have to perform ASR at first before the forced-alignment preprocessing and the proposed method can then be performed. I think this process is not so convenient.

---

### Note · Authors · 2024-11-14

I have read and agree with the venue's withdrawal policy on behalf of myself and my co-authors.